# Novel Chain-End Modification of Polymer Iodides via Reversible Complexation-Mediated Polymerization with Functionalized Radical Generation Agents

**DOI:** 10.3390/polym15122667

**Published:** 2023-06-13

**Authors:** Kazuya Ohtani, Kanta Shimizu, Tatsuhiro Takahashi, Masumi Takamura

**Affiliations:** 1Department of Organic Materials Science, Graduated School of Organic Materials Science, Yamagata University, 4-3-16 Jonan, Yonezawa 992-8510, Japan; t221002m@st.yamagata-u.ac.jp (K.O.); smknt19981007@gmail.com (K.S.); effort@yz.yamagata-u.ac.jp (T.T.); 2Yamagata University Inkjet Development Center, 1- 808-48 Arcadia, Yonezawa 992-0119, Japan

**Keywords:** reversible complexation-mediated polymerization, chain-end modification, living radical polymerization, radical generating agent, MALDI-TOF MS

## Abstract

The modification of polymer chain ends is important in order to produce highly functional polymers. A novel chain-end modification of polymer iodides (Polymer-I) via reversible complexation-mediated polymerization (RCMP) with different functionalized radical generation agents, such as azo compounds and organic peroxides, was developed. This reaction was comprehensively studied for three different polymers, i.e., poly (methyl methacrylate), polystyrene and poly (n-butyl acrylate) (PBA), two different functional azo compounds with aliphatic alkyl and carboxy groups, three different functional diacyl peroxides with aliphatic alkyl, aromatic, and carboxy groups, and one peroxydicarbonate with an aliphatic alkyl group. The reaction mechanism was probed using matrix-assisted laser desorption/ionization time-of-flight mass spectrometry (MALDI-TOF MS). The combination of PBA-I, iodine abstraction catalyst and different functional diacyl peroxides enabled higher chain-end modification to desired moieties from the diacyl peroxide. The dominant key factors for efficiency in this chain-end modification mechanism were the combination rate constant and the amount of radicals generated per unit of time.

## 1. Introduction

The modification of polymer chain-ends, in addition to main chains and side chains, is important in order to produce highly functional polymers. Fox et al. first investigated the relationship between the number of polymer chain-ends and the glass transition temperature (T_g_) and discovered that lower-molecular-weight polymers, which have more high-mobility chain-ends, have a lower T_g_ [1].

Since this discovery, many researchers have studied the effects of polymer chain-ends on physical properties. Polymer chain-ends are classified into two chemical moieties, i.e., non-functional and functional groups. For polymers with non-functional groups as chain-ends, surface orientation [2], self-organization [3], and intermolecular interactions [4,5] have been reported to improve the physical properties of the polymers. For polymers with functional chain-ends, so-called telechelic polymers, interfacial adhesion [6] and particle dispersibility [7] have been reported.

Among the methods to modify polymer chain-ends, living polymerization is one of the most effective since it enables not only the control of chain-end groups but also gives a narrow molecular weight distribution, allowing more uniform functionality. Reversible-deactivation radical polymerization (RDRP) with different reversible activation mechanisms has also been reported [8]. RDRP methods include nitroxide-mediated polymerization (NMP) [9] with a dissociation-combination (DC) mechanism, organotellurium-mediated living radical polymerization (TERP) [10] and reversible addition-fragmentation chain transfer polymerization (RAFT) [11] with degenerative chain transfer (DT) mechanisms, atom transfer radical polymerization (ATRP) [12] and reversible complexation-mediated polymerization (RCMP) [13] with atom transfer (AT) mechanisms, and reversible chain transfer catalyzed polymerization (RTCP) [14] with a reversible chain transfer (RT) mechanism.

In these RDRP systems, a functionalized alkyl group of the RDRP initiator is transferred to the α-position chain-end of the polymer, and a dormant group of the initiator is transferred to the ω-position chain-end. This dormant ω-position chain-end limits the applications of the resulting polymer due to toxicity, coloration, odor or low thermal stability. Therefore, many studies on the removal and modification of dormant chain-ends have been conducted, such as radical-induced reduction [15,16,17] using nucleophilic agents, photoinduction [18], thermal decomposition [19], and reduction [20]. For simultaneous removal and modification of dormant chain-ends, chain-end modification of thiol groups [21] and various functional groups [22] using nucleophilic substitution reactions, hydroxyl groups using an oxidizing agent [23], maleimide groups using radical addition-fragmentation coupling reactions [24] and azido and (meth) acryloyl groups [25] using a click reaction have been reported. Previous reports on the simultaneous removal and modification of dormant chain-ends showed highly efficient chain-end modification (more than about 90%) [21,23,24,25]. However, there are problems such as limitations on the functional groups that can be introduced [21,23] and side reactions such as cyclization [24]. Unfortunately, highly efficient modification of dormant ω-position chain-ends while introducing various moieties with no side reactions, such as cyclization, has rarely been obtained.

Herein, we propose a novel method to simultaneously remove and modify chain-end iodine in RCMP polymers, using radical generating agents such as azo compounds and organic peroxides.

## 2. Materials and Methods

### 2.1. Materials

As monomers, methyl methacrylate (MMA) [>98%, Fujifilm Wako Pure Chemical Co. (Wako), Osaka, Japan], styrene (St) (>99%, Wako) and n-butyl acrylate (BA) (>99%, Wako) were used after the removal of inhibitors by distillation.

As initiators and iodine abstraction catalysts for RCMP, 2-iodo-2-methylpropionitrile (CP-I) [>96%, Tokyo Chemical Industry (TCI), Tokyo, Japan], ethyl 2-iodo-2-phenylacetate (PhE-I) (>97%, TCI), tributylmethylphosphonium iodide (BMPI) (>98%, TCI), tetrabutylammonium iodide (BNI) (>98%, TCI), tetrabutylammonium bromide (BNBr) (>98%, TCI) and tributylsulfonium iodide (BSI) (>95%, TCI) were used without further purification.

As solvents, diethylene glycol dimethyl ether (diglyme) (>99%, TCI), dehydrated 1,4-dioxane (DOX) [>99.5%, Kanto Chemical Co., Inc. (Kanto), Tokyo, Japan], dehydrated toluene (toluene) (>99.5%, Kanto), acetone (>99%, TCI), tetrahydrofuran (THF) (>99%, Kanto), methanol (>99.8%, Wako), hexane (>99%, Wako), chloroform-d_1_ with 0.03% TMS (CDCl_3_) (>99.8%D, Kanto) and dimethyl sulfoxide-d_6_ (DMSO-d_6_) (>99.9%D, Kanto) were used without further purification.

As radical generating agents, azo compounds, 2.2′-azobis(2-methylpropionitrile) (AIBN) (>97%, Kanto) and 4,4′-azobis(4-cyanovaleric acid) (ACVA) (>98%, Wako), and organic peroxides, dilauroyl peroxide (LPO) [>98%, NOF Corporation (NOF), Tokyo, Japan], dibenzoyl peroxide (BPO) (>75%, NOF), disuccinic acid peroxide (SAPO) (>80%, NOF), and bis(4-tert-butylcyclohexyl) peroxydicarbonate (TCPO) (>90%, NOF), were used without further purification.

As a matrix and cationization agent for MALDI-TOF MS analysis, trans-2-[3-(4-tert-butylphenyl)-2-methyl-2-propanylidene] malononitrile (DCTB) (>98%, TCI) and sodium trifluoroacetate (NaTFA) (>98%, TCI), respectively, were used without further purification.

### 2.2. Preparation of Polymer-I Precursor via RCMP

The preparation of Polymer-I as a precursor via RCMP, including reagents, reaction conditions (temperature × time), and general procedures are shown in Figure 1. The chemical structures of initiators and iodine abstraction catalysts used in this study are summarized in Figure 1. All reactions were carried out by bulk polymerization in the same manner as described by Chen et al. [22]. A mixture of a monomer (40 mmol), an initiator (0.4 mmol of PhE-I for PMMA-I, 0.8 mmol of CP-I for PSt-I and 0.35 mmol of CP-I for PBA-I), a polymerization rate increasing agent (0.3 mmol of AIBN for PSt-I) and an iodine abstraction catalyst (0.4 mmol of BMPI for PMMA-I, 0.2 mmol of BNI for PSt-I and 0.94 mmol of BNI for PBA-I) was heated in a 30 mL Schlenk flask at a prescribed temperature for a prescribed time under a nitrogen atmosphere in dark conditions with magnetic stirring. The reaction was quenched by cooling to room temperature. The mixture was diluted by a solvent (acetone for PMMA-I, THF for PSt-I, and none for PBA-I), and Polymer-I was reprecipitated in a poor solvent (hexane for PMMA-I, methanol for PSt-I and water/methanol (9/1) (*vol*/*vol*) for PBA-I) and dried in a vacuum oven.

The iodine abstraction rate for Polymer-I by the iodine abstraction catalyst is highly dependent on the stability of the polymer radical after iodine abstraction. Good iodine abstraction temperatures have been reported to be around 70 °C for PMMA, around 80 °C for PSt, and around 110 °C for PBA [26].

### 2.3. Chain-End Modification of Polymer-I

The possible chain-end modification methods for Polymer-I with a radical generating agent are shown in Figure 2. In this scheme, chain-end modification can begin with two simultaneous reactions, i.e., the generation of a polymer radical triggered by iodine abstraction from Polymer-I, and the exchange reaction proposed by Chen et al. [22]. This is followed by a combination reaction of the resulting polymer radical with the functionalized radical generated by the thermal decomposition of the functionalized radical generating agent, which is our additional proposed mechanism. In the bottom reaction, *k_c_* represents the rate constant for the combination reaction between the polymer radical and the functionalized radical, while *k_d_* indicates the thermal decomposition rate constant for the radical generating agent [27], which affects the amount of radicals generated per unit time.

Chemical structures and one-hour half-life temperatures (T_1h_) for the radical-generating agents used in this study are summarized in Figure 2. These agents were used as model functional groups to obtain modified polymers with chain-ends that provide physical interactions/chemical bonding. Applications that utilize the hydrophilic-hydrophobic interaction of aliphatic and aromatic moieties are conceivable. For example, aliphatic and aromatic moieties can be used as adsorption sites for polymeric dispersants for pigments in an aqueous environment [28,29]. Introduction of reactive functional groups, such as carboxylic acid to the polymer chain-end results in telechelic polymers, that can be used as crosslinkers.

The general procedures for chain-end modification of Polymer-I are described below. All reactions were carried out under conditions in which the radical generating agents were more than 99% decomposed.

A mixture of Polymer-I (1 eq as 300 mg, 10 wt%), a radical generating agent (10 eq or 2 eq), an iodine abstraction catalyst (10 eq or 5 eq) and a solvent (90 wt%, DOX for PMMA-I, toluene for PS-I, toluene for PBA-I except when using SAPO, diglyme for PBA-I when using SAPO) was heated in a 30 mL Schlenk flask under a nitrogen atmosphere and dark conditions at a prescribed temperature and for a prescribed time with magnetic stirring. The reaction was quenched by cooling to room temperature.

Centrifugation (3000 rpm for 5 min) was performed for the removal of solid BNI, and the chain-end-modified polymer was reprecipitated from the supernatant by a poor solvent (hexane for chain-end modified PMMA, methanol/THF (8/2) (*vol*/*vol*) or none for chain-end modified PBA) and dried in a vacuum oven.

### 2.4. Experiment to Estimate the Combination Rate Constant k_c_

It is known that oxy radicals produced by the decomposition of peroxides are more reactive than carbon radicals produced by the decomposition of azo compounds. Thus, different radical-generating agents may lead to different rates for the combination reaction between radicals generated by the decomposition of the radical-generating agent and the polymer radicals generated by iodine abstraction, and this affects the chain-end modification ratio. Therefore, a model experiment was carried out to determine the rate constant *k_c_* for the combination reaction between polymer radicals generated by iodine abstraction from the low-molecular-weight alkyl iodide initiator PhE-I instead of Polymer-I with iodine abstraction catalyst BNI, and the radicals generated by decomposition of radical generating agents (azo compound AIBN, diacyl peroxides LPO and BPO). Furthermore, a ^1^H-NMR analysis was carried out to determine the combined ratios for radicals (R) generated by the thermal decomposition of radical generating agents (R-R) and PhE radicals (PhE) generated by iodine abstraction from alkyl iodide (PhE-I) by BNI.

A mixture of PhE-I (1 eq as 0.2 mmol), a radical generating agent (5 eq) decomposed at 100 °C for 1 h, and BNI (5 eq) was heated in a 30 mL Schlenk flask under a nitrogen atmosphere and dark conditions at 100 °C for 1 h with magnetic stirring. The reaction was quenched by cooling to room temperature. The radical generating agent was LPO (5 eq, 100% decomposition), BPO (6 eq, 83% decomposition), or AIBN (5 eq, 100% decomposition) at the same molar ratio, taking into account the decomposition ratio at 100 °C for 1 h using published decomposition parameters [27].

### 2.5. Characterization

To determine the molar mass, size exclusion chromatography (SEC; GPC SYSTEM-21, Shodex, Tokyo, Japan) was performed using a chromatograph equipped with a pump, three gel columns (GPC K-803, KF-806L and KF-804L, Shodex), a RI-detector (RI-71S, Shodex), and an absorbance detector (UV–8020, TOSOH Corporation, Tokyo, Japan), based on a conventional calibration curve using polystyrene standards. The eluent was tetrahydrofuran (THF) with a flow rate of 1.0 mL/min (40 °C).

To estimate the iodine chain-end removal ratio for Polymer-I, the chemical composition of chain-end modified polymers and the combination rate constant *k_c_*, ^1^H-NMR spectroscopy [JNM-EC500 (500 MHz), JNM-ECZ600R (600 MHz), JEOL, Tokyo, Japan] were used. CDCl_3_ and DMSO-d_6_ were used as solvents for the ^1^H-NMR analysis. Chemical shifts were calibrated using residual TMS (at 0.00 ppm) as the internal standard. Regarding the iodine chain-end removal ratio for Polymer-I, PMMA-I, PSt-I and PBA-I are identified as side-chain methyl protons in the MMA attached to iodide (3.73–3.77 ppm, *d’* in Appendix A, see Appendix A), main-chain methine protons in the St attached to iodide (4.48–4.75 ppm, *c’* in Appendix A, see Appendix A) and main-chain methine protons in the BA attached to iodide (4.28–4.36 ppm, *c’* in Appendix A, see Appendix A), respectively.

For qualitative and quantitative analysis of chain-end-modified polymers, matrix-assisted laser desorption ionization time-of-flight mass spectrometry (MALDI-TOF MS) analysis was performed on JMS-S3000 Linear TOF and Spiral TOF (JEOL Ltd., Japan) spectrometers at an acceleration voltage of 20 kV in positive linear mode and positive spiral mode, respectively. External mass calibration was performed using a PSt standard (Mn = 5000). DCTB was used as the matrix, and sodium trifluoroacetate as the cationization agent.

## 3. Results and Discussion

### 3.1. PMMA-I Modification

#### 3.1.1. Effect of Chemical Structure of Radical Generating Agent with or without Iodine Abstraction Catalyst on Chain-End Modification Efficiency

Table 1 shows the effect of the chemical structure of the radical generating agent with or without an iodine abstraction catalyst on the modified PMMA molar mass and the iodine chain-end removal ratio determined by ^1^H-NMR. For the molar mass distribution and the chemical composition before and after chain-end modification of PMMA-I, representative SEC chromatograms and ^1^H-NMR spectra are shown in Appendix A, see Appendix A, respectively. The reaction temperatures and times in Table 1 were determined as the conditions under which all the radical-generating agent was decomposed after the reaction. Thus, based on the T_1h_ shown in Figure 2, the reaction temperature and time that gave similar decomposition rates for radical generating agents were used. In order to obtain efficient chain-end iodine abstraction from the Polymer-I, the modification was carried out at temperatures close to 70 °C, which is the polymerization temperature.

##### Effect of Chemical Structure of Radical Generating Agent without Iodine Abstraction Catalyst on Chain-End Modification Efficiency

From the molar masses and molar mass distributions shown in Table 1 and Appendix A, see Appendix A, at the same radical generation rate during modification (T_1h_ + 10 °C, Entries 1–3), PMMA modified with TPCO (Entry 3) contains a small amount of high-molecular-weight PMMA (approximately twice as high as the main modified component), while PMMA modified with AIBN (Entry 1) or aliphatic LPO (Entry 2) has almost same molar mass distribution as PMMA-I. These results suggest there are two possible pathways involving higher-molecular-weight PMMA: either recombination of two PMMA radicals derived from iodine abstraction from PMMA-I, or inter-molecular crosslinking of two PMMA radicals derived from hydrogen abstraction of PMMA by peroxy radicals generated by decomposition of TCPO, which is the highest hydrogen abstraction ability among all type of organic peroxides.

As shown in Table 1, AIBN (Entry 1) gives a lower iodine chain-end removal ratio (21%) than the organic peroxides (Entry 2: 44% with LPO; Entry 3: 38% with TCPO peroxydicarbonate). In other words, more effective iodine chain-end removal is possible with organic peroxides than with azo compounds.

In the ^1^H-NMR spectra shown in Appendix A, see Appendix A, (Entries 1–3), no peaks derived from PMMA modified with the radical generating agent chain-end are found. This indicates that PMMA with the desired chain-end structure was not obtained. In addition, for PMMA modified with TCPO (Entry 3), a peak associated with -CH_2_–CH_2_– (*, 1.55–1.67 ppm) is observed. However, no peaks are seen to be associated with other PMMA structures produced by the side reactions, such as those involving recombination or disproportionation of two PMMA radicals due to iodine abstraction from PMMA-I or combination between PhMe radicals due to hydrogen abstraction from the solvent (methyl group of PhMe) by peroxy radicals generated by decomposition of TCPO and PMMA radicals due to iodine abstraction from PMMA-I. Taking into account the small amount of high-molecular-weight PMMA (Entry 3 in Appendix A, see Appendix A), this result may indicate intermolecular crosslinking of two PMMA radicals derived from hydrogen abstraction of methyl protons from the PMMA main chain by peroxy radicals generated by the decomposition of TCPO.

##### Effect of Addition of Iodine Abstraction Catalyst on Chain-End Modification Efficiency

From Table 1 and Appendix A, see Appendix A, PMMA modified by diacyl peroxides containing carboxy groups (Entry 4: SAPO in the absence of BNI, Entry 5: SAPO in the presence of BNI) for the same radical generation rate during modification (T_1h_ + 5 °C), the presence of BNI maintains the same molar mass distribution as PMMA-I, while the absence of BNI gives rise to a small amount of high-molecular-weight PMMA. In addition, from Table 1, the presence of BNI (Entry 5: 72%) gives a higher iodine chain-end removal ratio than the absence of BNI (Entry 4: 48%).

In Appendix A, see Appendix A, in the presence of BNI (Entry 5), a –CH_3_ peak (*c**, 1.67–1.75 ppm) and a –CH_2_– peak (*e*, 2.30–2.33 ppm) derived from PMMA modified with a SAPO moiety chain-end are observed in the ^1^H-NMR spectrum. However, no peaks associated with other PMMA structures, such as those produced by inter-molecular crosslinking of two PMMA radicals, recombination or disproportionation of two PMMA radicals, or combination between DOX (solvent) radicals and PMMA radicals, are observed. In contrast, in the absence of BNI (Entry 4), peaks (*, 1.55–1.67 ppm) indicating inter-molecular crosslinking of two PMMA radicals appear. Considering the iodine removal ratio, it can be concluded that Entry 5 contains PMMA modified with SAPO moiety chain-ends. This indicates that the iodine abstraction catalyst only abstracts iodine to generate PMMA radicals, and organic peroxides only generate radicals for coupling with PMMA radicals since the iodine abstraction catalyst abstracts iodine from PMMA-I faster than the generating radicals derived from the decomposition of organic peroxides in Entry 5.

##### Effect of Chemical Structure of Radical Generating Agent with Iodine AbstractionCatalyst on Chain-End Modification Efficiency

As shown in Table 1 and Appendix A, see Appendix A, both PMMA modified with SAPO (Entry 5) and with an azo compound containing a carboxy group (Entry 6: ACVN) in the presence of BNI at the same radical generation rate during modification (T_1h_ + 5 °C) maintain the same molar mass distribution as PMMA-I. From Table 1, PMMA modified with SAPO (Entry 5, 72%) gives a significantly higher iodine chain-end removal efficiency than that modified with ACVN (Entry 6: 33%). Furthermore, in Appendix A, see Appendix A, Entry 5 contains PMMA modified with a SAPO moiety chain-end, while Entry 6 does not include any particular peaks compared with PMMA-I. From these results, only PMMA modified with SAPO in the presence of BNI gives the desired PMMA modified with a SAPO moiety chain-end among all the modified PMMAs in this study.

In order to determine the chain-end modification ratio more directly, MALDI-TOF MS analysis was performed. The chain-end modification ratio for each polymer chain-end structure was roughly calculated by weight conversion of the area ratio for the corresponding peaks. Table 2 shows the polymer structures and chain-end modification ratios for PMMA modified with SAPO in the presence of BNI (Table 1, Entry 5) and its precursor PMMA-I.

While PMMA with iodine chain-ends was not observed, polymers P1 and P2 with different chain-end structures were observed. P1 is PMMA with an unsaturated C=C bond (theoretical mass = 7091.4, experimental mass = 7091.6, chain-end modification ratio = 63%) generated by the elimination of HI from PMMA-I. P2 is PMMA with a lactone ring structure (theoretical mass = 7077.4, experimental mass = 7078.6, chain-end modification ratio = 37%) generated by the elimination of CH_3_I from PMMA-I. The carbon−iodine bond of PMMA-I is so weak that it could be cleaved during the MALDI−TOF MS analysis. Thus, polymers P1 and P2 are considered to be generated from PMMA-I during the MALDI-TOF MS analysis [22,30].

For PMMA modified with SAPO in the presence of BNI (Entry 5), PMMAs with various chain-end structures were observed. P3 is PMMA with a SAPO moiety chain-end (theoretical mass = 7109.4, experimental mass = 7111.5, chain-end modification ratio = 30%), and P4 is PMMA with a hydrogen chain-end (theoretical mass = 7093.4, experimental mass = 7093.5, chain-end modification ratio = 10%). P5 is considered to be obtained by degradation of PMMA-I remaining after chain-end modification or removal of CH4 from P4 during the MALDI-TOF MS analysis (theoretical mass = 7076.4, experimental mass = 7078.5, chain-end modification ratio = 60%). This result indicates that PMMA modified with SAPO in the presence of BNI contains PMMA with at least 30% of chain-ends modified by a SAPO moiety.

In order to clarify the reason why PMMA modified with SAPO (Entry 5; diacyl peroxide) in the presence of BNI contains PMMA with at least 30% of chain-ends modified with the desired moiety, while PMMA modified with ACVN (Entry 6; azo compound) does not, the combination reaction rate constants *k_c_*, shown in Figure 2, are listed in Appendix A, see Appendix A. It can be seen that there is a significant difference in combination ratios between azo compound AIBN (18%) and diacyl peroxides LPO and BPO (100%). Therefore, since the oxy radicals derived from peroxide decomposition react with PMMA radicals faster than the carbon radicals derived from the azo compound, radicals derived from diacyl peroxide with higher *k_c_* are considered to be more effective than those derived from the azo compound with lower *k_c_*. Thus, the chain-end modification efficiency is strongly affected by the combination reaction rate constant *k_c_*, which is one of the dominant factors in Figure 2.

### 3.2. PSt-I Modification

#### Effect of Type of Iodine Abstraction Catalyst on Chain-End Modification Efficiency

Table 3 shows the effect of the type of iodine abstraction catalyst on the modified PS molar mass, iodine chain-end removal ratio determined by ^1^H-NMR and iodine chain-end modification ratio determined by MALDI-TOF MS. In terms of the molar mass distribution and the chemical composition before and after chain-end modification of PSt-Is, representative SEC chromatograms and ^1^H-NMR spectra are shown in Appendix A, see Appendix A, respectively.

Table 3 and Appendix A, see Appendix A, show that for the same radical generation rate during modification (T_1h_ + 0 °C, Entries 8–10), all PSts modified with iodine abstraction catalysts with different iodine abstraction ratios (Entries 8 and 8′: BNI; 74%; Entry 9: BNBr; 55%; Entry 10: BSI; 27%) [26], the lower-molecular-weight region was slightly extended while maintaining a monodisperse molecular weight distribution. From these results, it is assumed that there are possible pathways containing slightly higher-molecular-weight PSt formed by the recombination of two lower-molecular-weight PSt radicals derived from iodine abstraction from PSt-I because of faster diffusion of such radicals.

In Table 3, PSt modified with an iodine abstraction catalyst with a higher iodine abstraction rate (Entries 8 and 8′: BNI; 74%; Entry 9: BNBr; 55%) [26] gives a higher iodine chain-end removal ratio (100%) than that (34%) having a lower iodine abstraction rate (Entry 10: BSI; 27%) [26].

As shown in Entries 8–9 in Appendix A, see Appendix A, for iodine abstraction catalysts with a higher iodine abstraction rate (Entries 8 and 8′: BNI; 74%; Entry 9: BNBr; 55%) [26], three distinct peaks are observed for the desired PSt modified with LPO moiety chain-ends without decarboxylation. The peaks are associated with –CH– (*c**, 5.17–5.40 ppm), –CH_2_– (*e*, 1.23–1.28 ppm) and –CH_3_ (*f*, 0.89 ppm). In addition, the –CH– peak (*c*’, 4.48–4.75 ppm) associated with PSt-I is found to have disappeared. A peak associated with –(CPh)CH–CH(CPh)– (*, 2.97–3.18 ppm) is also observed. No peaks are observed that are associated with other PSt structures, such as those formed by disproportionation of two PSt radicals, a combination between PhMe (solvent) radicals and PSt radicals, or inter-molecular crosslinking of two PSt radicals derived from hydrogen abstraction of methyl protons from the PSt main chain or phenyl protons of PSt side chains by peroxy radicals generated by decomposition of LPO.

The slight extension of the lower-molecular-weight region in the SEC chromatograph in Appendix A, see Appendix A, while maintaining a monodisperse molecular weight distribution, may indicate that PSt derived from bimolecular termination of low-molecular-weight PSt chain-end radicals was generated. On the other hand, in Entry 5 (PSt modified with BSI with the lowest iodine abstraction rate of 27%) [26], the –CH– peak (*c*’, 4.48–4.75 ppm) identified as PSt-I and the –(CPh)CH–CH(CPh)– peak (*, 2.97–3.18 ppm) were observed. This suggests that a modification using a low iodine abstraction catalyst resulted in slow iodine chain-end abstraction and polymer radical generation, leaving a large amount of the precursor PSt-I (66%).

Figure 3 shows MALDI-TOF MS spectra of PSt modified with LPO in the presence of BNBr (Entry 9, which had the highest LPO moiety modification ratio) and its precursor PSt-I (Entry 7). Polymer peaks at regular intervals of 104 (St unit) are observed. In precursor, PSt-I (Entry 7), PSt with iodine chain-ends are not observed, but polymers P6–8 with different chain-end structures are observed. P6 is a PSt-macromonomer with an unsaturated C=C bond (theoretical mass = 3109.9, and experimental mass = 3109.6, chain-end modification ratio = 70%) generated by the elimination of HI from PSt-I. The carbon−iodine bond of PSt-I is so weak that it could be cleaved during the MALDI−TOF MS analysis. Thus, this PSt-macromonomer is considered to be generated from PSt-I during the MALDI-TOF MS analysis [31,32]. P7 is PSt with cyanoisopropyl moieties at both chain-ends (theoretical mass = 3178.9, experimental mass = 3178.6, chain-end modification ratio = 14%) generated by the bimolecular termination of two PSt chain-end radicals. The structure of P8 (chain-end modification ratio = 15%) is unidentified.

In PSt modified with BNBr (Entry 9), three polymer series (P9–11) were observed. P9 is a PSt with an LPO moiety chain-end (theoretical mass = 4559.8, experimental mass = 4559.5, chain-end modification ratio = 33%). P10 is PSt with a hydrogen chain-end (theoretical mass = 4569.8, experimental mass = 4569.2, chain-end modification ratio = 55%). P11 is a polymer with cyanoisopropyl moieties at both chain-ends (theoretical mass = 4636.8, experimental mass = 4636.3, chain-end modification ratio = 13%) generated by the bimolecular termination of two PSt chain-end radicals. P9–11 were also observed in a similar ratio in Entries 8 and 8’. In Entry 10, P6 or P10 (60%), P11 (11%) and an unidentified peak (29%) were observed. P11 is the polymer generated during the polymerization of Entry 7 and chain-end modification reaction for Entry 9.

The main chain-end structure for Entry 9 was P10. It is considered that the peak derived from P10 is not observed in Appendix A, see Appendix A, because this peak is thought to appear at the same position as peak *c*. Due to the fast iodine abstraction and the long lifetime of the chain-end radical, P10 could be generated by chain transfer of PSt chain-end radicals generated by iodine abstraction to the solvent or other reagents.

As seen for Entries 8–10, PSt modified with BNBr (Entry 9, 33%) gives a higher LPO moiety modification ratio than the control PSt modified with BNI (Entry 8′: 29%) while that with BSI (Entry 10) does not include any chain-end structure with the LPO moiety. The chain-end modification ratio for Entry 9 (33%) having an alkyl group is lower than that for PSt modified with an amine having an alkyl group (75%), as reported by Chen et al. [22].

From the SEC, ^1^H-NMR and MALDI-TOFMS measurements, PSt modified with LPO moiety chain-ends having no decarboxylation reaction was obtained:

### 3.3. PBA-I Modification

#### 3.3.1. Effect of Type of Diacyl Peroxide on Chain-End Modification Efficiency

Table 4 shows the effect of the type of diacyl peroxide on the modified PBA molar mass, the iodine chain-end removal ratio determined by ^1^H-NMR, and the iodine chain-end modification ratio determined by MALDI-TOF MS. Appendix A, see Appendix A, show SEC chromatograms and ^1^H-NMR spectra indicating the molar mass distribution and the chemical composition before and after the chain-end modification of PBA-Is.

Table 4 and Appendix A, see Appendix A, show that at the same temperature (110 °C, Entries 12–14), all PBAs modified with diacyl peroxides having different chemical structures (LPO for Entry 12, BPO for Entry 13, and SAPO for Entry 14) in the presence of BNI, no significant change of either molecular weight (Mn and Mw) or PDI including the molar mass distribution was observed after the reaction. This indicates that recombination or inter-molecular crosslinking of two PBA radicals does not occur.

Table 4 shows that PBAs modified with diacyl peroxides having non-polar groups (LPO for Entry 12 and BPO for Entry 13) exhibit a higher iodine chain-end removal ratio (100%) than that (83%) in the presence of a polar group (SAPO for Entry 14). In Appendix A, see Appendix A, for PBAs modified with diacyl peroxides having a non-polar group (LPO for Entry 12 and BPO for Entry 13), four distinct peaks are observed for Entry 12 for the desired polymer (PBA modified with LPO moiety chain-ends without decarboxylation). These peaks are associated with –CH– (*c**, 4.86–4.96 ppm), –CH_2_– (*h*, 2.32–2.36 ppm and *i*, 1.21–1.33 ppm) and -CH_3_ (*j*, 0.86–0.90 ppm). For Entry 13, three distinct peaks are observed for the desired polymer (PBA modified with BPO moiety chain-ends without decarboxylation). These peaks are associated with –CH– (*c***, 5.15–5.27 ppm) and –CH– in the benzene ring of the BPO moiety (*l + m*, 7.31–7.50 ppm), as well as the disappeared –CH– peak (*c*’, 4.28–4.36 ppm) identified as PBA-I. On the other hand, for PBAs modified with diacyl peroxides having a polar group (SAPO for Entry 14), three distinct peaks are also observed for the desired PBA modified with SAPO moiety chain-ends without decarboxylation. These peaks are associated with –CH– (*c****, 4.94–5.08 ppm) and –CH_2_– (*n*+*o*, 2.60–2.79 ppm), as well as a –CH– peak with reduced intensity (*c*’, 4.28–4.36 ppm) identified as PBA-I. No peaks are observed that are associated with other PBA structures, such as those formed by disproportionation of two PBA radicals or a combination between PhMe or diglyme (solvent) radicals and PBA radicals.

Figure 4 shows MALDI-TOF MS spectra of PBA modified with different diacyl peroxides and the precursor PBA-I (Entry 11). In all measurements (Entries 11–14), peaks at regular intervals of 128 (BA unit) were observed. In the spectrum of precursor PBA-I (Entry 11), only P12 with an iodine chain-end was observed. In PBA modified with LPO (Entry 12), only P13 with an LPO moiety chain-end (theoretical mass = 6570.3, experimental mass = 6570.7, chain-end modification ratio = 100%) was observed. In PBA modified with BPO (Entry 13), P14 with a BPO moiety chain-end (theoretical mass = 6570.3, experimental mass = 6570.7, chain-end modification ratio = 85%) and unidentified peaks were observed.

In PBA modified with SAPO (Entry 14), PBAs P12,15–18 with various chain-end structures were observed. P15 is PBA with the SAPO moiety chain-end (theoretical mass = 6488.2, experimental mass = 6488.8, chain-end modification ratio = 25%). P16 is PBA with an OH chain-end (theoretical mass = 6388.1, experimental mass = 6388.9, chain-end modification ratio = 19%) generated by hydrolysis of the SAPO moiety of P15. P17 is PBA with a lactone ring structure (theoretical mass = 6442.2, experimental mass = 6442.8, chain-end modification ratio = 23%). The structure of P18 is unidentified. Only peaks derived from PBA modified with SAPO, which is P15, and PBA-I, which is P12, are observed in Appendix A, see Appendix A. Therefore, P16 and P17 could have been generated by hydrolysis of the SAPO moiety of P15 and elimination of CH_3_(CH_2_)_3_I from PBA-I during the MALDI-TOF MS analysis, respectively.

PBA modified with LPO (Entry 12) in the presence of BNI gives the highest LPO moiety modification ratio (100%) compared to modified PMMA (Entry 5: 30%) and modified PSt (Entry 8: 27%) under identical conditions. The PBA radical (secondary radical) derived from iodine abstraction from PBA-I is less stable than both the tertiary radical generated by iodine abstraction from PMMA-I and the styryl radical generated by iodide abstraction from PSt. Therefore, PBA-I could undergo slower iodine abstraction and have a shorter lifetime for chain-end radicals than PSt-I and PMMA-I. This suggests that relatively slow iodine abstraction is important for a high chain-end modification ratio. The chain-end modification ratio for modified PBA having an alkyl group (Entry 12, 100%) is the same as that for PBA modified with an amine having an alkyl group (100%), as reported by Chen et al. [22], whose chain-end structure consists of an amine and a cyclic structure of BA. From the standpoint of the desired chain-end modification, Entry 12 is more efficient than PBA modified with the amine.

Furthermore, for PBA-I modified with different diacyl peroxides (LPO: aliphatic moiety; BPO: aromatic moiety; and SAPO: aliphatic moiety with carboxyl group), PBA modified with SAPO (Entry 14: 25%) gives a lower chain-end modification ratio than those with LPO (Entry 12: 100%) and BPO (Entry 13: 85%). In Entry 14, when SAPO was added dropwise to the PBA-I solution containing BNI, a rapid color change from clear to black was observed. This peculiar color change could indicate a side reaction between the acidic SAPO and the iodine of BNI. These results suggest that the structure of the diacyl peroxide affects the chain-end modification ratio, which will be discussed later.

#### 3.3.2. Effect of Dose of Radical Generating Agents on Chain-End Modification Efficiency

Following the highly efficient chain-end modification of PBA in Entry 12, the effect of small amounts of radical generating agent on chain-end modification was investigated.

Table 5 shows the effect of the dose of radical generating agents on the modified PBA molar mass, the iodine chain-end removal ratio determined by ^1^H-NMR, and the iodine chain-end modification ratio determined by MALDI-TOF MS. Representative SEC chromatograms and ^1^H-NMR spectra are shown in Appendix A and Appendix A, see Appendix A, respectively, indicating the molar mass distribution and the chemical composition before and after chain-end modification of PBA-I.

Table 5 and Appendix A, see Appendix A, show that at the same temperature (110 °C, Entries 12, 15 and 16), for PBAs modified with different doses of LPO (5 eq for PBA-I in Entry 15 and 2 eq for PAB-I in Entry 16) in the presence of BNI, no significant change in either molecular weight (Mn and Mw) or PDI including the molar mass distribution was observed after the reaction. This indicates that recombination or inter-molecular crosslinking of two PBA radicals did not occur. For Entries 15 and 16 in Table 5, all PBAs exhibit a high iodine chain-end removal ratio (100%). Entries 15 and 16 are for PBA modified with LPO moiety chain-ends. No peaks associated with other PBA structures, such as those formed by disproportionation of two PBA radicals or a combination between PhMe or diglyme (solvent) radicals and PBA radicals, are observed. In the MALDI-TOFMS spectra of modified PBAs (Entries 15 and 16), only P13 with LPO moiety chain-ends was observed. A high chain-end modification ratio (100%) was also obtained for PBAs modified with a lower dose of LPO (5 eq for PBA-I in Entry 15 and 2 eq for PAB-I in Entry 16) than Entry 12 (10 eq of LPO for PBA-I). This indicates that highly efficient chain-end modification of PBA-I is possible with small doses of the radical-generating agents.

#### 3.3.3. Effect of Concentration of Radicals Generated by Decomposition of DiacylPeroxides

The chain-end modification efficiency for PBA modified with LPO (Entry 12: 100% at 110 °C) is higher than that for PBA modified with BPO (Entry 13: 85% at 110 °C). Since LPO has a lower T_1h_ than BPO, LPO decomposes faster than BPO at 110 °C. It is considered that the higher radical concentration for LPO gives a higher modification efficiency when the polymer radical is generated by BNI abstraction. Different radical concentrations for LPO are brought about by different *k_d_* values in Figure 2. In order to clarify the effect of *k_d_* for the radical generating agent on the chain-end modification efficiency, the relationship between *k_d_* and the chain-end modification ratio was investigated. Figure 5 shows the relationship between the logarithm of *k_d_* for the radical generating agent at the modification temperature and the chain-end modification ratio for the modified PBAs of several diacyl peroxides (LPO and BPO) and BNI (Entries 12 and 13). Entries 12 and 13 using LPO and BPO with the same *k_c_* shown in Appendix A, see Appendix A, and relatively high initiator efficiencies were compared. All modifications being compared have the same molar ratio of the radical generator to Polymer-I (10 eq for Polymer-I). Furthermore, since chain-end modification is performed at the same temperature (110 °C) and using the same solvent (dehydrated toluene), the concentration of radicals generated after all the radical generating agents are decomposed is also equal (20 eq for polymer radical). A proportional relationship between log *k_d_* and the chain-end modification ratio can be seen. Therefore, in the modification of PBA, the chain-end modification efficiency is thought to be affected by the radical concentration per time triggered by the decomposition rate constant *k_d_*, which is one of the dominant factors in Figure 2.

## 4. Conclusions

A novel chain-end modification method was demonstrated for several types of Polymer-I (PMMA-I, PSt-I and PBA-I) via RCMP with different functionalized radical generation agents such as azo compounds and organic peroxides, which enabled the introduction of the desired moieties from the radical generating agents. The combination of PBA-I, iodine abstraction catalyst and different functional diacyl peroxides gave higher chain-end modification to the desired moiety. The key factors for obtaining higher modification efficiency in this chain-end modification mechanism were discussed.

## Data Availability

The raw data presented in this study are available on request from the corresponding author.

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
