# Peer review of "Novel Chain-End Modification of Polymer Iodides via Reversible Complexation-Mediated Polymerization with Functionalized Radical Generation Agents"

_polymers, 2023, doi:10.3390/polym15122667_

Round 1

Reviewer 1 Report

This paper should not be accepted for publication in its present form.  No information is provided regarding experimental reproducibility, the results are presented without explaining why experimental conditions are chosen, and the structures assigned by ESI-MS analysis are not explained (and also are two units away from theoretical m/z values).  Moreover, the yields of the modification process are quite low, except for the case of PBA-I. Thus, while the method may be considered novel, it does not seem any more successful than other procedures listed in the introduction (lines 56-61). 

Perhaps the manuscript can be improved to be suitable for publication after re-writing to clearly present results and support the mechanistic interpretations.  However, more experiments may also be required, as the following complete list of issues must be addressed:   

1.      (pg 1, line 28-30, 1st paragraph) It is incorrect to claim that the influence of polymer MW on Tg was discovered only in 1997 (reference 1), as the knowledge dates back to Fox and Flory in 1950.

2.      Pg 1, line 41-43, 3rd paragraph) It is misleading to say that living radical polymerization techniques “have been reported recently”, as the first publications in the area were from the early 1990s.  Moreover, the IUPAC naming of this chemistry as Reversible-Deactivation Radical Polymerization (RDRP) should be used, as it is not correct to refer to these methods as living, since radical-radical termination does occur.

3.     Scheme 1 indicates AIBN was a component in PSt-I synthesis, but was not used for synthesis of PMMA-I or PBA-I (even though the latter structure is shown with an AIBN-chain end). These differences are not consistent with the stated experimental procedures, a problem which must be corrected.

4.     Section 2.4 describes a procedure to determine the “combination rate constant”, kc by mixing the initiator PhE-I with the radical generator,  measuring the products formed by NMR.  Is it expected that this rate coefficient also is representative of combination of the polymer species (e.g., PMMA-I) with primary radicals formed by decomposition of the radical generator (such as LPO)? Please justify this assumption, as it is well-known that radical-radical termination rates are controlled by diffusion, and thus related to the lengths of the two terminating species.

5.     It is not clear how the reaction temperatures were selected for the Table 1 set of experiments. Were these chosen to provide roughly the same radical generation rate for the experiments (based on the half-life temperatures reported in Figure 1)? This important point needs to be discussed.

6.      What is the experimental uncertainty associated with the values in Table 1? Is there a significant difference between 44 and 48% iodine chain-end removal?  Or is this difference smaller than those observed if one experiment is repeated with multiple trials?  (Similarly, the 27 vs 29% Table 3 result is not a convincing difference to discuss, without any discussion of experimental uncertainty.)  

7.     Has the lactone ring structure for P2 in Table 2 been reported before as a product of PMMA-I chain-end degradation? If so, please use references 27 and 28 to directly support this observation. If not, further discussion of this mechanism (through formation of CH3-I) is required.

8.     Elimination of O2 to form P5 from P3 (Table 2 structures) seems like a difficult mechanism. The difference of 2 units between theoretical and expected m/z values for P3 and P5 are much higher than most ESI analysis.  Is there any supporting evidence from model-chemistry reactions (e.g., PhE-I + SAPO) to validate these structures? What are the mechanisms proposed?

9.     A temperature of 90 C was used for LPO combined with PMMA-I (Table 1), but only 80 C for LPO combined with PST-I (Table 2).  In addition, an iodine abstraction catalyst was used in every case with PST, but not used for PMMA.  These differences are not explained and make it very difficult to compare the two polymer systems. It is not correct to claim that Entry 5 in Table 1 and Entry 8 in Table 2 can be compared, as the two experiments use different radical generators, different ratios of the polymer and radical generator, and were conducted at different temperatures (and thus different rates of radical generation).  

10.  Why was 110 C used for PBA-I experiments (Table 4)? While I understand that the reaction temperature may need to be modified based on stability of the polymer-iodine bond, it also has a profound influence on the generation rate of radicals (and thus radical concentration) from the radical-generator (e.g., LPO), which I think would have a significant influence on the chain-end modification ratio.  Please consider/discuss this important point.

11.   Why are iodine chain-ends with PBA observed by ESI-MS (Figure 4), but not the other polymers?

12.   (Pg 12, line 356-361) This comparison is not valid, as Entry 5 in Table 1 is not with LPO.  Indeed the previous sentence correctly says that Entry 5 is with SAPO.  Moreover, the LPO experiment for PMMA-I with LPO (Entry 2) did not employ BNI as co-catalyst, unlike with the experiments with PST-I (Entry 8) and PBA-I (Entry 12).

13.   A key point is made in lines 361-362 (pg 12), that the reaction temperature was modified for PST and PBA experiments compared to PMMA to maintain the same iodide abstraction/recombination rate constant.   This important information should be conveyed to the reader before presenting the experimental results; otherwise too many questions arise as the reader goes through the “Results” section.  Literature values for ka and kda (with references) should be supplied to support this statement.

14.   In Scheme 3, why is not the termination of the R radical with the I-catalyst radical considered?  I also suggest that Scheme 3 be presented before the discussion/presentation of experimental results, as it is needed to understand the choice of experimental conditions.

15.   Given the Scheme 3 mechanism, why were most of the experiments with PMMA-I (summarized in Table 1) conducted without any iodine abstraction catalyst present in the system?

16.   Given the result that higher temperature (thus higher kd) leads to increased chain-end modification ratio, wouldn’t it be expected that LPO with PMMA-I reacted at >90  C in the presence of BNI would improve the results? Why was this experiment not conducted?

17.   The chain-end modification reaction (scheme 2) was conducted under dilute conditions (with 90 wt% solvent) and with relatively high reagent concentrations (5-10 eq of iodine abstraction catalyst; 2 or 10 eq of radical generator). In addition, it only works efficiently with P-BA, as the chain-end modification ratios for PS and PMMA are quite low.  Can this really be considered an efficient process of interest that can be used for all polymer types?

Reviewer 2 Report

I reviewed the manuscript “Novel Chain-end Modification of Polymer-Iodides via Reversible Complexation Mediated Polymerization with Functionalized Radical Generation Agents “ by Kazuya Ohtani, Kanta Shimizu, Tatsuhiro Takahashi, and Masumi Takamura.

The manuscript describes the study on polymer chain-end modification via radical reaction. The topic is highly relevant in the scope of functional materials synthesis as it broadens the possibilities to obtain highly functionalized tailor made polymers.

Even though I find this manuscript quite interesting and highly-relevant, I cannot recommend its publication in the present form.

Prior to publication, the authors should address the following:

1.       Justification. When giving the introduction and discussing the end-functionalization of polymers, the authors didn’t mention the yields of modification which are typical for these types of reactions. Thus it is hard to evaluate if the results obtained by the authors align with the literature data, close any gap or demonstrate a breakthrough. The authors didn’t mention why they need the particular functional groups which they study in the current research. After such a modification, the polymer chain loses its ability to initiate the polymerization further on (i.e. the “living” character) without attaining some useful molecular or material properties. So the purpose of this chemistry remains unclear to me.

2.       Experimental design. The authors used radical chemistry to conduct the modification of the polymer end-chain groups. It is well-known that a lot of side reactions can take place during radical reactions involving polymer chains: chain transfer to polymer, migration of the free radical center, chain scission after migration. These reactions are highly relevant for polyacrylates at temperatures above 333K. In this scope, the result of 100% modification of polyacrylate is surprising to me, especially taking into account long reaction time (9h) and high temperature (110 deg C).

I strongly suggest that the authors discuss the possible side reactions during modification and try to evaluate their importance in the described conditions.

3.       Characterization. In my opinion, the authors can put more effort into the characterization of the synthesized polymers. They show MALDI-TOF results only which by themselves are not always conclusive. I strongly suggest that the authors study the NMR of the polymers they synthesized (both 1H and 13C), and provide the overlays of the SEC data.

4.       What are the error margins of modification yields estimation? In Figure 2 (1H NMR) I see a lot of resonances with similar chemical shifts. It means that the integration accuracy is low with possible overestimations of yields. Quadruplet with a chemical shift of 3.8 ppm is unidentified in the spectrum whereas it overlaps with the signal of interest d’. Clearly, some new NMR resonances appear in the spectrum after modification: singlet at δ=3.4 ppm, doublet and triplet at δ around 1.2 ppm. The authors didn’t discuss this in the manuscript whereas it can be important for product identification.

5.       I disagree with the approach to correlate the degree of chain-end modification and the rate coefficient of radical initiator decomposition! First of all, the rate of the reaction is proportional to both, the rate coefficient and concentration of the reagent. Secondly, the radical initiators are known to have so-called “efficiency”: they tend to undergo side reactions. Thus the number of radicals generated upon initiator decomposition is smaller than the number of initiator molecules decomposed *2. Next, the diffusion can play a significant role for these types of reactions.

Some minor comments:

-        The text should be thoroughly proofread and corrected. I see a lot of expressions from the spoken language which is unacceptable in scientific publications: “make sense”, “there is-there are”, etc.

-          Some figure captions need to be cleaned up.

Spoken language should be avoided.

Round 2

Reviewer 1 Report

The modifications made greatly improve the paper; everything is now clear and well-explained.

Good, minor (and few) errors

Author Response

Dear Reviewers

Thank you for your valuable suggestions on our manuscript.

We reconfirmed the SEC chromatographs for determining the molecular weight distribution for the modified polymer before and after the chain-end modification reaction, and also 1H-NMR spectra to clarify the main (side) reactions during chain-end modification. To illustrate the characteristic changes observed, representative SEC chromatographs (Figure S4 for PMMA, Figure S6 for PSt, Figure S8 for PBA) and 1H-NMR spectra (Figure S5 for PMMA, Figure S7 for PSt, Figure S9 for PBA) for each polymer are provided in the Supplementary Material. Based on the results obtained by SEC, 1H-NMR and MALDI analysis, the products obtained by chain-end modification were confirmed and are explained in the relevant sections of the manuscript.

Reviewer 2 Report

I received the response letter and the corrected manuscript by Kazuya Ohtani, Kanta Shimizu, Tatsuhiro Takahashi, and Masumi Takamura. Despite the authors putting a lot of effort into correcting the manuscript, I still cannot recommend publishing it in its present form.

11.       Considering side reactions and characterization techniques. From one point of view, the authors claim that no side reaction exists in the described systems as the Mn, Mw and Đ remain almost the same. On the other, their NMR spectra show a lot of unidentified signals which is the result of some unaccounted reactions. The authors even admit that “Other polymers (P16, P17, P19 and P20) which are thought to be formed by side reactions were also observed, but the details are still unclear.” Thus, I see a deep contradiction here which is essential for the results in general.

22.       The authors claim that “significant change of either molecular weight (Mn and Mw) or PDI did not occur … ”, but the changes caused by the side reactions can be subtle. That is why I asked to provide the overlay of two SEC traces to ensure that indeed all the changes are minor.

33.       When discussing the side reactions, the authors claim that chain transfer to solvent occurs. What is the reaction pathway for the solvent radical in this case?

44.       For the list of side reactions, the authors didn’t mention the H-abstraction of the initiator radical from the middle of the chain.

55.       Considering 99% of iodine chain-end removal ratios for all modified PSts and PBAs. When I look at the NMR spectra in Figure 3 I see that the decrease of the integral intensity of signal d’ is way below 99%. Furthermore, it has a complicated shape (a singlet that overlaps with a quadruplet at δ=3.8ppm). Thus it is impossible to integrate it precisely with 5% error which the authors claim.

66.       Table 4, entries 11 and 12. The authors claim 100% modification in this case. It would be very convincing if the authors provide NMR and SEC data as SI. If I compare entries 11 and 12, I see that the dispersity increase in entry 12 (Đ = 6.2/7.3 = 1.1774 ≈1.18) instead of 1.16 as the authors report. So I start to wonder about the accuracy of SEC data integration as well.

77.       Considering the chemistry which authors observe. When peroxyl-type initiators decompose, two consequent reactions occur: (1) O-O bond decomposition with formation of peroxy benzoyl radical, (2) de-carbonylation of peroxy benzoyl radical with formation of CO and benzoyl radical. [https://doi.org/10.1016/j.jhazmat.2008.03.078] The initiation of polymerization in bulk occurs with the peroxyl radical. But as soon as the solvent involved, the de-carbonylation becomes the dominant process. But the authors don’t observe the adducts of Ph radical with the polymer chain. I wonder what is the reason for that.

One of the possible explanations is so-called “chain transfer to initiator reaction” which authors didn’t discuss. [https://doi.org/10.1016/B978-0-08-096701-1.00070-7]

88.       Considering discussion on the kinetics: “All the three modifications being compared (Entries 8, 12, 13) have the same molar ratio of radical generator to Polymer-I (10 eq for Polymer-I). Furthermore, since chain-end modification is performed at a temperature at which all radical generating agents are decomposed, the concentration of radicals generated after all the radical generating agents are decomposed is also equal (20 eq for Polymer radical).” I emphasize once again: when 100% of radical generating agent decomposes, it doesn’t mean that 2 eq. of radicals are generated. The efficiency of initiation depends on the chemical structure of the initiator, temperature, solvent. Temperatures are different in the reaction conditions which authors describe. Thus it is impossible to compare. I strongly advise the authors not to make far-fetched conclusions about reaction kinetics here.

99.       The authors claim that the proofread the manuscript. But I read the updated version and see more misprints like the following example in the footnote of table 6:“a combinaacombination ratio between R· and PhE· was calculated by integral of peak corresponding to PhE-R generated by combination of PhE· and R·.

Proofreading required.

Round 3

Reviewer 2 Report

The authors answered all my comments. In my opinion, the manuscript can be published.

Minor revision is required.